

# *Microcos magnifica* (Sparrmanniaceae) a new species of cloudforest tree from Cameroon

Martin Cheek

Identification & Naming, Science, Royal Botanic Gardens, Kew

## ABSTRACT

**Background**. Although many new species to science have been discovered from thousands of specimens resulting from botanical inventories to support conservation management in Cameroon in recent years, additional species remain to be formally evaluated taxonomically and described. These include species from genera which have been taxonomically neglected for many decades in Africa, such as *Microcos*.

**Methods**. This study is based mainly on herbarium specimens and field observations made in Cameroon during a series of botanical surveys. Herbarium material was examined with a Leica Wild M8 dissecting binocular microscope fitted with an eyepiece graticule.

**Principal Findings**. *Microcos magnifica* Cheek (*Malvaceae-Grewioideae* or *Sparrmanniaceae*) is described as an Endangered (EN B2 ab(iii)) new tree species from the submontane forests of Cameroon. It is illustrated and described, and its conservation status and taxonomic affinities are assessed. It is the first new *Microcos* described from Africa in more than 90 years and is unique on the continent in having sculptured fruits.

**Discussion**. A systematic revision, with a molecular phylogenetic study, of *Microcos* Burm. ex L. in Africa is necessary if the affinities of the species, including *M. magnifica*, are to be reliably established.

## INTRODUCTION

During identification of specimens resulting from botanical surveys of Mt Kupe and the Bakossi Mts in SW Region, Cameroon, specimens of a remarkable undescribed *Microcos* Burm. ex L. (*Linnaeus, 1753*) came to light, which were designated as *Microcos* sp. A (Cheek in *Cheek et al., 2004*: 414). Subsequently an additional specimen was discovered in the forests of Ebo, Littoral Region. Here these specimens are formally named as *Microcos magnifica* Cheek, the first new species to science to be described in the genus for Africa for 90 years. The species is remarkable for its sculptured fruit surfaces which are verrucate. Sculptured fruit surfaces are not otherwise known in the African species but do occur in some Asian species.

*Microcos* is a palaeotropical genus of about 77 species (http://powo.science.kew.org/taxon/urn:lsid:ipni.org:names:328157-2) based on *M. paniculata* Burm. ex L. (1753) from Sri Lanka. *Linnaeus (1767)* later synonymised *Microcos* under *Grewia* L. However, the genus was resurrected by *Burret (1926)*.

Corresponding author
Martin Cheek, m.cheek@kew.org

Burret's authoritative revision (1926) of former Tiliaceae sens. lat. presaged its break-up into todays's Brownloideae/Brownlowiaceae, Tiliaceae sensu stricto/Tilioideae and Grewioideae/Sparrmanniaceae (with the largest number of genera and species) including *Microcos* (*Bayer et al., 1999*; *Bayer & Kubitzki, 2003*; *Cheek, 2007a*; *Cheek, 2007b*; *Cheek, 2007c*). In addition, *Schoutenia* Korth.included by Burret in Tiliaceae, is now placed in Dombeyoideae/Pentapetaceae (*Cheek, 2007d*). Burret's was the last global treatment of *Microcos* (1926). He recognised 53 species, of which 19 were recorded from Africa and 34 in Asia.

Of the 99 names in *Microcos* listed in http://ipni.org/, http://powo.science.kew.org/taxon/urn:lsid:ipni.org:names:328157-2 accept 77 names. The majority of these 77 are in S.E. Asia, but with 10 in Africa. The genus is absent from the Neotropics and Madagascar.

Illogically, while *Microcos* has been maintained as a separate genus from *Grewia* in Asia (e.g., *Chung, 2003*; *Chung, 2006*; *Chung, Soepadmo & Lim, 2005a*; *Chung & Soepadmo, 2011*), the two genera have often been united under *Grewia* in Africa. For example, in one of the most recent Flora accounts of *Grewia* (including *Microcos*) for Africa, *Whitehouse (2001)* states " …Kirkup followed Burret in recognising *Microcos* as a distinct genus; this concept has also been followed in SE Asia. Although there are clear differences between *Microcos* and the other sections of *Grewia*, for consistency I am following the practice set by the other African floras, of not recognising.…"

This practice is maintained widely today, for example by the excellent and essential African Plant Database at the Conservatoire et Jardin botaniques de la Ville de Genève (http://www.ville-ge.ch/musinfo/bd/cjb/africa/).

In fact the two genera are readily recognised as expressed in the key below, modified from that in *Whitehouse (2001)*:

Trees and climbers, rarely shrubs, of evergreen forest; stigmas entire; fruit unlobed; inflorescences terminal, sometimes axillary also, many-flowered .............. ***Microcos***

Shrubs, rarely trees, of bushland or woodland; stigmas lobed; fruit 4-lobed, rarely entire; inflorescences usually axillary or leaf-opposed, rarely terminal, usually few-flowered ......................................................................................................... ***Grewia***

According to the molecular analysis of *Brunken & Muellner (2012)*, *Microcos* is not embedded in *Grewia*, neither are they sister groups, and they fall into distinct clades.

Additional characters for separating the two genera are found in the pollen, wood anatomy and in the leaf anatomy, particularly the epidermal cells (*Chattaway, 1934*; *Chung, 2002*; *Chung, Soepadmo & Lim, 2003*; *Chung et al., 2005b*). *Microcos* was maintained in *Bayer & Kubitzki (2003)*.

The genus *Microcos* has been little studied in Africa, as evidenced by the fact that the first new name in African *Microcos* since 1926 was published in 2004 (*Microcos barombiensis* (K. Schum.) Cheek in *Cheek et al., 2004*: 414). In the course of matching the material described as new in this paper, it became clear that a revision of the genus for Africa is desirable to address specimen misidentifications and additional apparently undescribed species. It is hoped to address these problems in a future paper.

## MATERIALS & METHODS

The electronic version of this article in Portable Document Format (PDF) will represent a published work according to the International Code of Nomenclature for algae, fungi, and plants (ICN), and hence the new names contained in the electronic version are effectively published under that Code from the electronic edition alone. In addition, new names contained in this work which have been issued with identifiers by IPNI will eventually be made available to the Global Names Index. The IPNI LSIDs can be resolved and the associated information viewed through any standard web browser by appending the LSID contained in this publication to the prefix "http://ipni.org/". The online version of this work is archived and available from the following digital repositories: PeerJ, PubMed Central, and CLOCKSS.

This study is based mainly on herbarium specimens and field observations made in Cameroon during a series of botanical surveys beginning in 1991. These surveys were mainly led by the author. So far they have resulted in 52,450 specimens being studied at K and YA, of which 37,850 were newly collected, the data stored on the Kew Cameroon specimen Access database (Gosline, p. 11 in *Cheek et al., 2004*). The top set of specimens was initially deposited at SCA, and later YA, duplicates being sent to K. The fieldwork was approved by the Institutional Review Board of the Royal Botanic Gardens, Kew entitled the Overseas Fieldwork Committee (OFC). The most the most recent OFC approval is numbered 807. The most recent invitation to effect research on the flora and vegetation of Cameroon has the reference number 050/IRAD/DG/CRRA-NK/SSRB-HN/09/2016. It is issued under the terms of the five year Memorandum of Collaboration between Institute for Research in Agricultural Development (IRAD)-Herbier National du Cameroun and Royal Botanic Gardens, Kew signed 5th Sept 2014.

All specimens cited have been seen by the author unless indicated n.v. Herbarium citations follow Index Herbariorum (http:// sweetgum.nybg.org/ih/) and binomial authorities (http://ipni.org/). Material of the suspected new species was compared morphologically with material of all other African *Microcos* (or *Grewia* sect. *Microcos* (L.)Wight and Arnott) principally at K, but also using material from WAG. This comprised about 350 specimens.The online search address used for retrieving specimen data from labels at P was http://coldb.mnhn.fr/catalognumber/mnhn/p/p00375109. Burret's types of *Microcos* at B were destroyed by allied bombing in 1943 so it was not possible to consult them. This has necessitated that subsequent authors select neotypes of his names, e.g., *Whitehouse (2001)*. The description follows the format of *Whitehouse (2001)*.

The conservation assessment was made using the categories and criteria of *IUCN (2012)*. The extent of occurrence was calculated with Geocat (*Bachman et al., 2011*). Herbarium material was examined with a Leica Wild M8 dissecting binocular microscope. This was fitted with an eyepiece graticule measuring in units of 0.025 mm at maximum magnification. The drawing was made with the same equipment using Leica 308700 camera lucida attachment.

## RESULTS

### Key to the tree species of *Microcos* in Africa west of Democratic republic of Congo & the Congo river

1.  Leaves deeply toothed. Maiombe Mts of Cabinda ................. *M. gossweileri* **Burret**
    Leaves entire. Nigeria to Congo-Brazzaville, but unknown from Cabinda ............. 2

2.  Leaf base cuneate, leaf surfaces glabrous; fruits glossy, smooth ...............................
    ................................................................................................. *M. coriacea* **Burret**

    Leaf base truncate or cordate, leaf blade lower surface stellate hairy; fruits matt,
    verrucate .................................................................................. *M. magnifica* **Cheek**

*Microcos magnifica* *Cheek* species novum
Holotype: Cameroon, S.W. Province, Mt Kupe, Kupe village, main trail towards summit, fr. 9 July 1996, *Etuge* 2886 (holo. K; isotypes BR, K, MO, P, SCA, US, WAG, YA) (Figs. 1 and 2)
*Microcos* sp. A, Cheek (in *Cheek et al., 2004*: 414).

Tree 20–35 m tall, 30–70 cm diameter at breast height, crown small, bole straight, base of bole with 4–5 concave slender buttresses reaching 1 to 1.5 m above the ground where sometimes spreading up to 2.5 m from the trunk and branching.

Bark dull medium red–brown, fibrous; slash hard fibrous-granular, without scent or exudates, white, oxidising rapidly from white to red.

Leafy stems 3–5 mm diameter below the third node, finely longitudinally ridged, densely minutely grey–brown puberulent, internodes 2.5 cm long.

Leaves obovate, obovate-oblong or elliptic, 12.5–25.5 × 6.6–13.5 cm (those of sterile stems large, to 28 cm long ), acumen 0.4–1.8 cm long, base truncate or truncate and abruptly cordate, margin entire, lateral nerves 11–13 on each side of the midrib, the basal pair more conspicuous by virtue of a pair subsidiary nerves, brochidodromous domatia absent, tertiary nerves strongly scalariform, quaternary nerves inconspicuous: upper surface with midrib varied, convex, densely and minutely grey-brown puberulent, secondary nerves flat but also puberulent: lower surface with midrib and secondary nerves strongly raised, brownish green, the areolae pale green or brown/khaki densely puberulent with minute pale brown 8–20-armed stellate hairs 0.1–0.2 mm diameter, touching each other, more or less completely concealing the epidermis. Presumed shade leaves (larger, from sterile branches—*Elad* 118) with hairs sparse, separated by 1 or 2 hair diameters, smaller, 0.075–0.1 mm diameter, with only 6—8 (–12) arms. Petiole stout, cylindrical, (1.5−)1.8–2 ×0.3 cm. Stipules caducous, not seen, but leaving an arched scar 4 mm long on the stem each side and 1 mm below the insertion of the leaf base.

Inflorescence and flowers unknown. Infructescence terminal, paniculate, 11–16 × 5.5–13 cm, bearing 5–13(–12) fruits; peduncle 1.5–2.7 cm; bracts not seen; pedicel absent, fruits articulated at junction with stem.

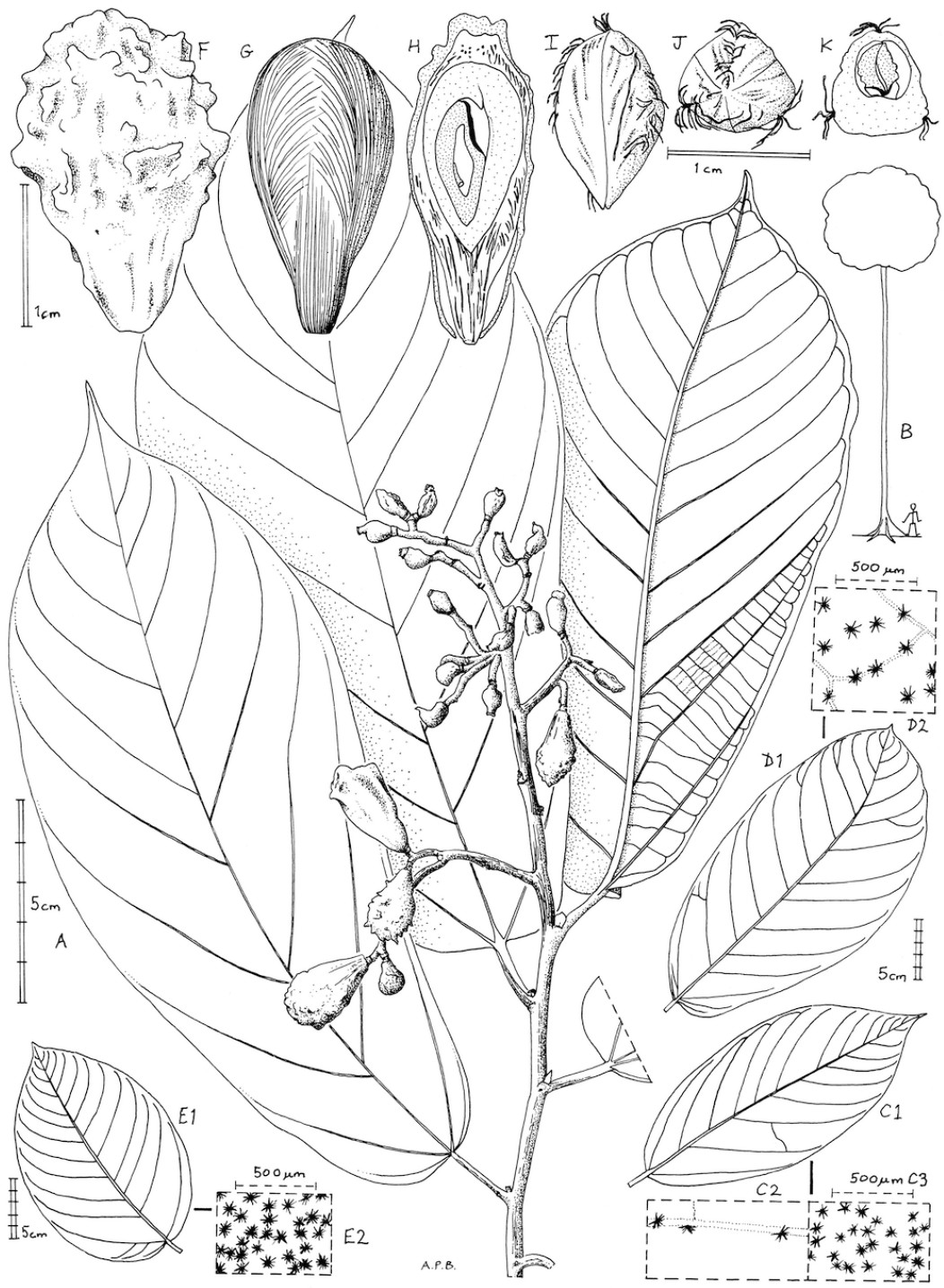

**Figure 1** *Microcos magnifica.* (A) habit, fruiting branch; (B) habit sketch; (C) leaf variation: Etuge 2686: hairs on upper surface (left); hairs on lower surface (right). (D) leaf variation: Elad 118, with detail of hairs on lower surface; (E) leaf variation: Cable 2806, with detail of hairs on lower surface; (F) fruit, side view; (G) fruit, left with pericarp removed exposing mesocarp fibres; right longitudinal section (endocarp stippled, endosperm densely stippled), (H) endocarp, left, side view; right, distal end view; (I) endocarp with seed, transverse section. (A, C, F–K) from Etuge 2686; (B) from field observations of Cheek 12980; (D) Elad 128; (E) Cable 2806. All drawn by Andrew Brown.

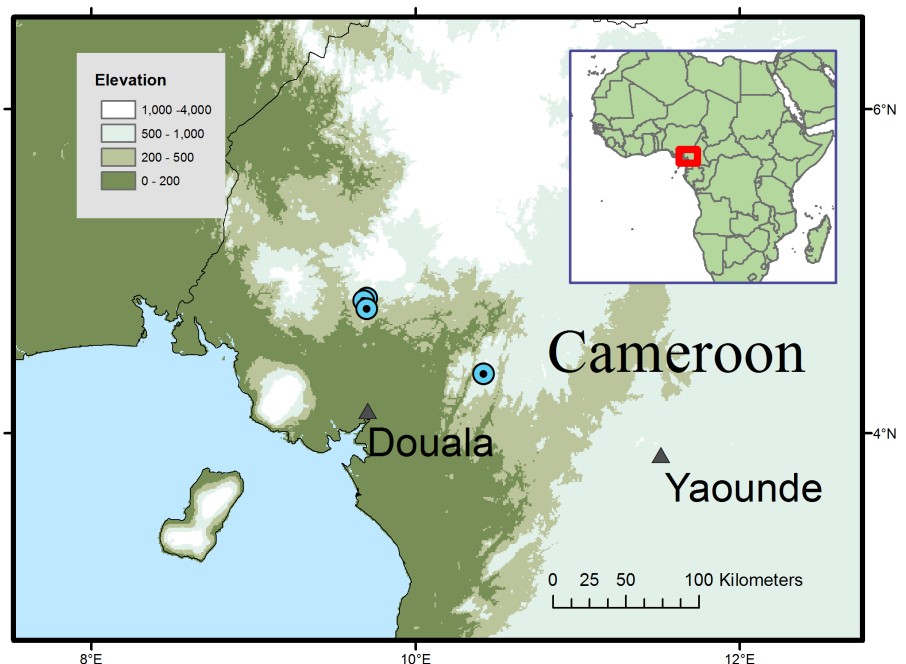

**Figure 2** Global distribution of *Microcos magnifica*.

Fruits fleshy, red when live, drying pink-brown, obovoid to ellipsoid 2–2.4 × 1.2–1.5 cm, verrucate and finely longitudinally wrinkled with 20–25 verrucae, verrucae 0.5–1 mm long, patent. Mesocarp: outer part thin and fleshy, inner part thick and densely fibrous. Endocarp obovoid, slightly 3-angled, woody, whitish brown, sutures longitudinal, alternating with three lines of hairs; locule 1, probably by abortion from 3, 1-seeded.

Seed narrowly ovoid, glabrous, slightly laterally compressed, hilum subapical; endosperm extensive, embryo flattened.

**Phenology:** Fruiting: April to July; flowering: unknown.

**Distribution and habitat:** SW and Littoral Regions of Cameroon; submontane or submontane-lowland forest with *Medusandra mpomiana* Letouzey & Satabie, *Santiria trimera* (Oliv.) Aubrév., *Allanblackia gabonensis* (Pellegr.) Bamps, *Coelocaryon preussii* Warb. (Mt. Kupe); *Pycnanthus* Warb., *Coelocaryon* Warb. *Staudtia* Warb., *Petersianthus* Merr., *Strombosia* Blume and *Maesobotrya* Benth. (Ebo); 750–1,000 m alt.

**Etymology:** Meaning magnificent, for the spectacular and unusual fruit ornamentation.

**Affinity:** Resembling *Microcos coriacea* Burret, but fruits verrucate and matt, not smooth and glossy; leaves with base truncate or cordate, not cuneate; lower surface densely white stellate hairy, not glabrous.

**Additional specimens:** South West Region. Mt Kupe, 15 Km WNW de Tombel, colline 930 m de NW de Ngussi, fr. 21 April 1976, *Letouzey* 14669 (P n.v.; YA 3 sheets); Mt Kupe, Nyasoso, trails above village, 4°49′N; 9°41′E, st. 6 Feb. 1995, *Elad* 118 (K, YA n.v.); Nyasoso, Max's trail, fr. 3 June 1996, *Cable* 2806 (K, YA); Kupe village, main trail towards summit, fr. 9 July 1996, *Etuge* 2686 (holo. K; iso. BR, K, MO, P, SCA, US, WAG, YA).

Littoral Region, Yingui, Ebo proposed National Park, 6 h walk S. of Iboti village; between the abandoned villages of Bekob and Masseng, 4 21 50 N; 10 25 20 E, st. 16 Feb. 2006, *Cheek* 12980 (K, SCA, YA)

**Conservation:** *Microcos magnifica* is here assessed as Endangered (EN B2 ab(iii)) using the IUCN 2012 system, since it is known from four threat-based locations with an extent of occurrence of 303 km$^2$ calculated using Geocat (*Bachman et al., 2011*) and an area of occupancy of 16 km$^2$ using IUCN preferred 4 km$^2$ grid cells. The species is threatened at all its known locations, most immediately the three in the Mt Kupe area of Ngussi and Nyassoso where clearance of forest continues upslope from the volcanic, fertile lowlands of the Chide valley. The clearance is for small-holder agriculture, principally for food crops. The locations concerned are all far outside the Mt Kupe Ecological Reserve and on the edges of towns. It is quite likely that some or all of the trees that provided the specimens and the forest remnants in which they occurred, have been cleared already (M Cheek, pers. obs., 2007). In order to reduce the threat to the species here, a local conservation poster featuring the species is intended in order to raise awareness of the existence and importance of its protection. However, at the fourth location, in the proposed Ebo National Park, the species is secure from immediate threat, there being no resident human population. However, the future of Ebo as a protected area is not certain, and logging, plantation and mining are all threatened as alternative uses for the land.

Since there is no indication that more than a single mature individual has ever been recorded at each of the four locations, it is conceivable that *Microcos magnifica* might be better assessed as Critically Endangered under Criterion D of IUCN (less than 50 mature individuals recorded).

## DISCUSSION

The affinities of *M. magnifica* may be with the only two other arborescent species of *Microcos* that occur in West-Central Africa (see key to species above). The majority of *Microcos* species in Africa are scandent climbers, completely different in habit from the arborescent species. Of the arborescent species, only *M. coriacea* is sympatric with *M. magnifica*. At Mt Kupe the two species have differing altitudinal ranges, *M. coriacea* with a range of 200–420 m, based on four records, and *M. magnifica* with 900–1,000 m, based on three records (Cheek in *Cheek et al., 2004*: 414). It can be postulated that *M. magnifica* has arisen as a submontane derivative of *M. coriacea*. However, among the taxa discovered as new at Mt Kupe at similar altitudes to *M. magnifica* was *Kupea martinetugei* Cheek (*Cheek, Williams & Etuge, 2003*) which has its sister species in the Eastern Arc Mts of Tanzania (*Cheek, 2004*).

Exactly the same geographic range as *Microcos magnifica*, which extends disjunctly from the western slopes of Mt Kupe to the submontane N-S ridge of the Ebo forest, is seen also in *Uvariopsis submontana* Kenfack (*Kenfack et al., 2003*) and *Costus kupensis* H. Maas and Maas (*Maas-van de Kamer et al., 2016*). It is remarkable that none of these conspicuous species has been discovered in the submontane ridge of the Ngovayang massif to the east at Bipindi, nor in the Bakossi Mts, immediately West of Mt Kupe, despite significant botanical surveys in these areas by Zenker and by K-YA teams respectively. This suggests that these

distributions are real and not the result of undercollecting. However, all of these species are infrequent and only known from three to six specimens in as many locations.

The discovery of such a distinctive new species in the Kupe-Bakossi, Ebo and adjoining areas is not unusual. Among other species discovered here were (in alphabetical order by genus): *Allophylus ujori* Cheek (*Cheek & Etuge, 2009a*), *Ancistrocladus grandiflorus* Cheek (*Cheek, 2000*), *Brachystephanus kupeensis* Champl. (*Champluvier & Darbyshire, 2009*), *Chassalia laikomensis* Cheek (*Cheek & Csiba, 2000*), *Coffea montekupensis* Stoff. (*Stoffelen et al., 1997*), *Coffea bakossii* Cheek & Bridson (*Cheek, Csiba & Bridson, 2002*), *Cola metallica* Cheek (*Cheek, 2002*), *Coleochloa domensis* Muasya & D.A. Simpson (*Muasya et al., 2010*), *Deinbollia oreophila* Cheek (*Cheek & Etuge, 2009b*), *Diospyros kupensis* Gosline (*Gosline & Cheek, 1998*); *Dovyalis cameroonensis* Cheek & Ngolan (*Cheek & Ngolan, 2007*), *Dracaena kupensis* Mwachala, Cheek, Eb. Fisch. et al. (*Mwachala et al., 2007*), *Impatiens etindensis* Cheek & Eb. Fisch. (*Cheek & Fischer, 1999*), *Impatiens frithii* Cheek (*Cheek & Csiba, 2002b*), *Isoglossa dispersa* I. Darbysh. (*Darbyshire, Pearce & Banks, 2011*), *Kupea martinetugei* Cheek & S.A.Williams (*Cheek, Williams & Etuge, 2003*), *Ledermanniella onanae* Cheek (*Cheek, 2003*), *Ledermanniella pollardiana* Cheek & Ameka (*Cheek & Ameka, 2008*), *Memecylon kupeanum* R.D.Stone, Ghogue & Cheek (*Stone, Ghogue & Cheek, 2008*), *Mussaenda epiphytica* Cheek (*Cheek, 2009*), *Newtonia duncanthomasii* Mackinder & Cheek (*Mackinder & Cheek, 2003*), *Oxyanthus okuensis* Cheek & Sonké (*Cheek & Sonké, 2000*), *Psychotria darwiniana* Cheek (*Cheek, Corcoran & Horwath, 2009*), *Psychotria geophylax* Cheek & Sonké and *P. bakossiensis Cheek & Sonké (2005)*, *Psychotria kupensis* Cheek (*Cheek, Horwath & Haynes, 2008*), *Psychotria moseskemei* Cheek (*Cheek & Csiba, 2002a*), *Rhaptopetalum geophylax* Cheek & Gosline (*Cheek, Gosline & Csiba, 2002*) and *Ternstroemia cameroonensis* Cheek (*Cheek, Tchiengue & Tacham, 2017*).

Most of these species are threatened with extinction, since they are narrow endemics with small ranges, restricted to mainly submontane (cloud) forest which is steadily being cleared, mainly for small-scale cultivation of food crops. These species feature in the Red Data Book of Cameroon (*Onana & Cheek, 2011*).

## CONCLUSIONS

A systematic revision, with a molecular phylogenetic study, of *Microcos* in Africa is necessary if the affinities of the species, including *M. magnifica*, are to be reliably established.

## ACKNOWLEDGEMENTS

Bate Oben is thanked for assisting by making morphological measurements. Xander van der Burgt assisted digitising the image. George Gosline is thanked for producing the map, and Saba Rokni for providing the Geocat analysis. Gaston Achoundong, Jean-Michel Onana, former heads of the National Herbarium of Cameroon (YA) and their successor Marie Florence Ngo Ngwe are thanked for their collaboration and support of the Royal Botanic Gardens, Kew's activities over the years. Laurence J. Dorr and two other, anonymous, reviewers are thanked for their constructive comments on an earlier version of this manuscript.

### Funding

Darwin Initiative grants 15034 and 8038 supported this research. In addition, fieldwork for the research was supported by the Earthwatch Institute 1993–2005. The author's salary during the study was paid by RBG, Kew. There was no additional external funding received for this study. The funders had no role in study design, data collection and analysis, decision to publish, or preparation of the manuscript.

### Grant Disclosures

The following grant information was disclosed by the author:
Earthwatch Institute 1993–2005: 15034, 8038.

### Competing Interests

The author declares there are no competing interests.

### Author Contributions

- Martin Cheek conceived and designed the experiments, performed the experiments, analyzed the data, contributed reagents/materials/analysis tools, wrote the paper, reviewed drafts of the paper.

### Ethics

The following information was supplied relating to ethical approvals (i.e., approving body and any reference numbers):

The fieldwork was approved by the Institutional Review Board of the Royal Botanic Gardens, Kew entitled the Overseas Fieldwork Committee (OFC).

### Field Study Permissions

The following information was supplied relating to field study approvals (i.e., approving body and any reference numbers):

IRAD-Herbier National du Cameroun sanctioned the field work under a series of Memoranda of Collaboration with the Royal Botanic Gardens, Kew, the most recent signed 5th Sept. 2014, extending five years from that date.

### Data Availability

The specimens on which this manuscript is based are housed in the herbaria for which the standard codes are K and YA. Specimen data and images of type material will be made available on the Kew Herbarium Catalogue at http://apps.kew.org/herbcat/gotoSearchPage.do.

### New Species Registration

The following information was supplied regarding the registration of a newly described species:

*Microcos magnifica* Cheek LSID: 77167392-1.

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
