# Peer review of "Microcos magnifica (Sparrmanniaceae) a new species of cloudforest tree from Cameroon"

_PeerJ, doi:10.7717/peerj.4137_

## Round 0.1 · original submission · Minor Revisions

· Academic Editor

Minor Revisions

Dear Martin,

The manuscript had a very good acceptance by the reviewers. Although the also found some points to be improved.

Please follow the suggestions by the reviewers to update and improve your manuscript.

Cheers,

Marcial.

Reviewer 1 ·

Basic reporting

New species are significant scientific discoveries, and it is important that such findings be published (and for new species to be properly described and named). This reviewer finds that such work is within the stated scope of the journal PeerJ.

The structure of the manuscript is consistent with that seen in short articles in the field of plant taxonomy. The structure also seems to be acceptable according to PeerJ standards.

The author of the manuscript is a first-language English speaker, but this reviewer finds that the way the text is written may be difficult for non first-language English speakers to understand. It almost seems as if the text was written in a hurry. Let me recommend that long sentences should be shortened and the writing should be done in a more direct (telegraphic) style.

The line drawing (Fig. 1) by Andrew Brown is fine work, but the version seen by this reviewer appears to be an early draft (not a finished plate). The version presented is not acceptable for publication.

The distributional map (Map 1) is also fine, but it would help to orientate the readers (especially those unfamiliar with the geography of Cameroon) if some additional landmarks could be added, e.g., administrative boundaries (regional and international) as well as the locations of major cities and/or regional capitals.

It would be really helpful if high-resolution scanned images of the specimens cited could be posted on the Kew Herbarium Catalogue (http://apps.kew.org/herbcat/gotoHomePage.do ) or a similar on-line database. This way the specimen-images and associated information would be publicly accessible.

Below are references to some additional, seemingly relevant literature which the author may want to cite in the Introduction of the revised manuscript:

Brunken U, & Muellner AH. 2012. A new tribal classification of Grewioideae (Malvaceae) based on morphological and molecular phylogenetic evidence. Systematic Botany 37: 699-711. [Grewia and Microcos as distinct genera, placed in different subclades of tribe Grewieae Endl. The ‘Microcos subclade’ also includes Colona Cav., Goethalsia Pittier, Luehea Willd. and Lueheopsis Burret.]

Chattaway MM. 1934. Anatomical evidence that Grewia and Microcos are distinct genera. Tropical Woods 38: 9-11.

Czarnecka E, Wiland-Szymańska J, Gawrońska K. 2006. Phytogeography of the genus Microcos L. (Malvaceae, Grewioidae) in Africa. Biodiversity Research and Conservation 3-4: 269-271.

Nurul-Aini CAC, Noraini T, Latiff A, Chung RCK, Nurhanim MN, Ruzi M. 2013. Systematic significance of petiole anatomical characteristics in Microcos L. (Malvaceae: Grewioideae). Malayan Nature Journal 65 (2&3): 145-170.

Shokefun EO, Ayodele,AE & Akinloye AJ. 2016. Systematic importance of leaf anatomical characters in some species of Microcos Linn. section Eumicrocos Burret in Nigeria. American Journal of Plant Sciences 7: 108-117. http://dx.doi.org/10.4236/ajps.2016.71012

Sprague TA. 1909. The section Microcos of Grewia in Africa. Bulletin of Miscellaneous Information (Royal Botanic Gardens Kew) 1909 (2): 66-68.

Experimental design

Fine although it would be good for the reader to know approximately how many specimens of African Microcos were studied by the author in the K, P and WAG herbaria as part of the pattern-matching exercise leading to the conclusion that the plants from Mt Kupe (Cameroon) are a new species.

This reviewer tested the hyperlink provided by the author for his search of African Microcos in the herbarium of MNHN Paris, and the resulting page was not of Microcos or Malvaceae sensu lato but was instead for a specimen-record of Ledermanniella schlechteri (Podostemaceae). This hyperlink thus needs to be double-checked and corrected if necessary.

Given that the last worldwide treatment of Microcos was published by Burret (1926), that Burret was based at the Botanischer Garten und Botanisches Museum Berlin-Dahlem, and that the Berlin herbarium was largely destroyed during WW2, it would be interesting for the reader to know to what extent our knowledge & further study of African Microcos is hampered by the type specimens having been lost. To what extent are Burret's types still extant or duplicated in other herbaria? Why was the Berlin herbarium evidently not consulted in the course of this study?

Validity of the findings

The author’s ‘Key to the species of Microcos tree species in Africa west of DRC & the Congo River’ includes only three species and is thus evidently incomplete. For example, the recent article by Shokefun et al. (2016) reported there are 8 species of Microcos in West Africa, and they reportedly sampled six of these species for their leaf anatomical study of Nigerian Microcos. The “missing” species in the present author’s key would thus seem to include at least M. africana, M. barombiensis, M. iodocarpa, M. malacocarpa, M. oligoneura and perhaps others as well.

Although I have no reason to doubt the author’s taxonomic judgement, the apparent omission of several West African species from the author's key leads this reviewer to think that with further checking the plants from Mt Kupe (Cameroon), proposed by the author as a new species M. magnifica, might prove to be a species previously described by a different author and under a different species name.

Additional comments

It is acknowledged there are differences of opinion on whether the major clades within the order Malvales should be treated as subfamilies within a broadly circumscribed Malvaceae or whether it is better to treat them as separate families. In fact the choice of rank for these taxa is more-or-less arbitrary. However, in the interest of avoiding any possible confusion, this reviewer recommends that just one family-name should be used in the TITLE of the manuscript, i.e. the family-name should be given as either Sparrmanniaceae or Malvaceae-Grewioideae, but not both. Of course it would still be appropriate to present some background information on these alternative family treatments in the INTRODUCTION.

·

Basic reporting

There are some inconsistencies in citations and in formatting in the references, but these are minor issues.

Experimental design

The manuscript presents a description of a new species from West Africa. It appears to satisfy all of the requirements set out in the International Code of Nomenclature for Algae, Fungi, and Plants (2012).

Validity of the findings

The author has made a convincing argument that this is indeed a new species and he has placed it in context of other taxonomic and conservation research in West Africa.

Reviewer 3 ·

Basic reporting

The paper provided interesting information on the new species of an endangered tree discovered in Cameroon, Africa. The abstract of this paper was adequately summarised the contents and the keywords were well representative of the paper. The introduction, results, discussion and references were clear, adequately discussed and cited.

Experimental design

In general the it was excellent and clearly written. The author appears to comply with the provisions of the ICN (McNeill et al., 2012) that are necessary for validly publishing a new species. Was this paragraph (Lines 132-142) necessary to be included in this section or could it be shortened the paragraph by just citing provisions in ICN.

Validity of the findings

The results are reasonable given as there were no inflorescences and flowers material since 2004 from all the reported locations.

Additional comments

*Lines 1 & 62: Delete this family (or Sparrmanniaceae) in the title and abstract. A synonym of Malvaceae sensu lato. Follow the Angiosperm Phylogeny Group IV (2016).
*Line 88-91: Author should include the subfam. Dombeyoideae as the genus Schoutenia from the ‘former Tiliaceae’ was placed in this subfamily. The article of Cheek (2007) was written for the family Sparrmanniaceae only. Better to cite Heywood et al. (2007) which was included other families such as Brownlowiaceae, Tiliaceae s.s. and Pentapetaceae.
*Line 108: Was this key to genera modified by the author based on his additional observations on both the genera.
*Line 111: Yes, in some Grewia species but it is not in Microcos.
*Line 150: Incomplete sentence, hanging.
*Line 176: The word “Verrucate” was used in the introduction and species description.
*Lines 297-298: Would the author will also consider to study the molecular phylogenetic of both Grewia and Microcos in Africa in order to establish the generic delimitation of these two genera for Africa.
**Map: Author should have a general map of Africa to show the location of Cameroon, then enlarge with this map to indicate the distribution of the new species.
***Other comments and suggestions to improve the paper, please refer to the attached PDF file.

Annotated reviews are not available for download in order to protect the identity of reviewers who chose to remain anonymous.

---

## Round 0.2 · Minor Revisions

· Academic Editor

Minor Revisions

Dear Martin,

Please find attached the comments from reviewers. Please, take into account those suggestions and submit a new version for our consideration for publication in PeerJ.

Cheers,

Marcial.

Reviewer 1 ·

Basic reporting

New species are significant scientific discoveries, and it is important that such findings be published (and for new species to be properly described and named). This reviewer finds that such work is within the stated scope of the journal PeerJ.

The structure of the manuscript is consistent with that seen in short articles in the field of plant taxonomy. The structure also seems to be acceptable according to PeerJ standards.

This reviewer suggests that all generic and species names where first mentioned in the main text should include the relevant authority for that name. The authority for the genus Microcos should also be cited in the Abstract, unless this would go against the journal’s editorial policy. As it now stands, there is one generic name (Schoutenia) and several species names (e.g., the names listed under the subheading ‘Distribution and habitat’) that are provided without authority. The generic name Microcos is mentioned in the first sentence of the Introduction, but the authority for this name doesn’t appear until the second paragraph; this problem could be solved by moving the authority to the first mention of the name Microcos in the first paragraph, or possibly by rearranging the paragraphs of the Introduction so that the background info about the genus appears first, with the specifics about the discovery of the new species Microcos magnifica coming later. There are two species names (Microcos gossweileri and Microcos coriacea) that first appear in the key to species, but without authority; in the case of the name Microcos coriacea, the authority is provided where the name is mentioned again later in the text, but in the case of Microcos gossweileri the key is the only place where this name is mentioned. The main point is that authorities for names are important and that these authorities should be mentioned consistently in the revised manuscript.

Experimental design

No comment.

Validity of the findings

In a previous version of this manuscript, the present reviewer pointed out that the author’s ‘Key to the species of Microcos tree species in Africa west of DRC & the Congo River’ includes only three species and is thus evidently incomplete. For example, the recent article by Shokefun et al. (2016) reported there are 8 species of Microcos in West Africa, and they reportedly sampled six of these species for their leaf anatomical study of Nigerian Microcos. The ‘missing’ species in the present author’s key would thus seem to include at least M. africana, M. barombiensis, M. iodocarpa, M. malacocarpa, M. oligoneura and perhaps others as well.

To this comment, the author has responded that ‘The key presented is to the tree species of Microcos in West-Central Africa, and not to the climbing species which include those mentioned by the paper referred to’.

This response is accepted, but this reviewer further notes it would be good for the author to mention explicitly the fact that there are additional Microcos species in West & Central Africa that are climbers, not trees (& perhaps state approximately how many species).

Additional comments

Once these minor, added comments have been addressed, then this reviewer recommends that the paper should be accepted for publication.

·

Basic reporting

The author addressed the concerns of the reviewers and the revised manuscript sontinues to satisfy the requirements of the International Code of Nomenclature for the publication of a new species name.

Experimental design

The experimental design is appropriate for the subject.

Validity of the findings

The conclusions seem appropriate.

Additional comments

There are two unresolved bibliographic citation issues -- missing journal number and questionable publication date. The attached version includes copy editing of the References especially as these were checked carefully against the manuscript.

Reviewer 3 ·

Basic reporting

This paper provides a new endangered tree species discovered from West Africa. The abstract, keywords, introduction, results, discussion and references are clear, adequately discussed and cited in the paper. This work is within the stated scope of the journal PeerJ.

Experimental design

It is excellent and clearly written. It follows the provisions of the ICN (McNeill et al., 2012).

Validity of the findings

The results are reasonable given by the author in the paper that this is indeed a new species in West Africa.

Additional comments

The author should follow APG IV (2016 in Bot. J. Linn. Soc. Vol. 181: 1-20), Mabberley (2017 in Mabberley's Plant-Book) and Steven's APG website (http://www.mobot.org/MOBOT/research/APweb/) to use family Malvaceae-Grewioideae which is widely accepted by all botanical journals. Sparrmanniaceae is a synonym in Malvaceae s.l. now as mentioned in Mabberley (2017) and also in Steven's APG website. The novel dismemberment of Malvaceae by Cheek (2006) and Cheek in Heywood et al. (2007) was not well-accepted by the botanical community/journals. In addition, the Heywood et al (2007) publication was not widely distributed and had not updated since 2007 as compared to the Steven's APG website and Mabberley (2017).

Other comments and suggestions to improve the paper, kindly refer to the attached PDF file.

Annotated reviews are not available for download in order to protect the identity of reviewers who chose to remain anonymous.

---

## Round 0.3 · accepted · Accept

· Academic Editor

Accept

Dear Martin,

I am glad to inform you that your study entitled " Microcos magnifica (Sparrmanniaceae) a new species of cloudforest tree from Cameroon" has been accepted for publication in PeerJ.
Congratultions!

Cheers,

Marcial.